# Semi-Empirical Pseudopotential Method for Graphene and Graphene Nanoribbons

**DOI:** 10.3390/nano13142066

**Published:** 2023-07-13

**Authors:** Raj Kumar Paudel, Chung-Yuan Ren, Yia-Chung Chang

**Affiliations:** 1Research Center for Applied Sciences, Academia Sinica, Taipei 11529, Taiwan; rajupdl6@gate.sinica.edu.tw; 2Molecular Science and Technology, Taiwan International Graduate Program, Academia Sinica, Taipei 11529, Taiwan; 3Department of Physics, National Central University, Chungli 320, Taiwan; 4Department of Physics, National Kaohsiung Normal University, Kaohsiung 824, Taiwan; cyren@nknu.edu.tw; 5Department of Physics, National Cheng-Kung University, Tainan 701, Taiwan

**Keywords:** semi-empirical pseudopotential (SEP), graphene, graphene nanoribbon, armchair graphene nanoribbon (AGNRs), density-functional theory (DFT), band structure, 2D materials, B-spline, mixed basis

## Abstract

We implemented a semi-empirical pseudopotential (SEP) method for calculating the band structures of graphene and graphene nanoribbons. The basis functions adopted are two-dimensional plane waves multiplied by several B-spline functions along the perpendicular direction. The SEP includes both local and non-local terms, which were parametrized to fit relevant quantities obtained from the first-principles calculations based on the density-functional theory (DFT). With only a handful of parameters, we were able to reproduce the full band structure of graphene obtained by DFT with a negligible difference. Our method is simple to use and much more efficient than the DFT calculation. We then applied this SEP method to calculate the band structures of graphene nanoribbons. By adding a simple correction term to the local pseudopotentials on the edges of the nanoribbon (which mimics the effect caused by edge creation), we again obtained band structures of the armchair nanoribbon fairly close to the results obtained by DFT. Our approach allows the simulation of optical and transport properties of realistic nanodevices made of graphene nanoribbons with very little computation effort.

## 1. Introduction

The electronic properties of low-dimensional materials are unique due to unprecedented properties that are unparalleled by those in bulk counterparts [1,2]. A scientific breakthrough occurred in 2004 in the isolation of monolayer graphene by mechanical exfoliation. After the discovery of graphene, the research on two-dimensional (2D) materials stimulated a great deal of interest due to their promising optical, physical, and chemical properties. These materials are layered, weakly coupled materials that can exist in a few or single-layer forms. Many 2D materials can be easily fabricated due to recently developed cutting-edge technology.

Ab initio formalism such as density-functional theory (DFT) [3] has been used extensively in calculating the electronic structures [4] and thermoelectric [5] and optical properties of solids [6]. In most calculations, three-dimensional (3D) plane waves are used as the basis functions. Plane wave (PW) basis is easy to implement, and the convergence of the calculation can be checked systematically. DFT methods are still affected by the band-gap problem, which requires numerically expensive GW calculation [7] or hybrid functionals to correct the band gap and effective mass near the Fermi level. The other drawback of this basis is the requirement of a large number of PWs needed when the unit cell is large.

Li et al. [8,9,10] introduced a planar-basis method to utilize plane waves along periodic (*x*-*y* plane) and Gaussian functions along a non-periodic (z) direction for ab initio calculations. The planar basis has advantages over the conventional PW basis in that it resumes the layer-like local geometry that appears in surfaces or 2D materials. Moreover, one can calculate the total energy for an isolated slab instead of using a superlattice of alternating slab and vacuum regions. Thus, the work function can be easily obtained [8]. The planar-basis method was further improved by Ren et al. [11,12] via the use of B-spline functions instead of Gaussian functions along the non-periodic direction. B-splines are not associated with atomic positions, making the geometry optimization easy to use, and the relevant matrices are sparse. Another advantage of using a planar basis is the ease of handling charged 2D materials such as gated 2D materials. This can avoid the artificial long-range Coulomb interaction introduced by the supercell method [13].

To study bulk materials with one or few atoms per unit cell, the empirical pseudopotential method (EPM) [14,15] presents an efficient and accurate method. EPM is extremely powerful since very few parameters are needed in order to obtain information about many properties of the solid, such as band structure, optical properties, and dielectric properties. Unlike DFT, EPM does not involve a self-consistent solution of the Schrödinger equation for charge density, thus greatly reducing time and computational cost. Furthermore, this method has a great advantage when working with a large unit cell of many identical atoms since the full potential of each atom can be calculated and fitted with a smaller number of parameters. In EPM, the pseudopotential contains both local and non-local pseudopotentials. The latter depends upon the angular momentum and energy. The incorporation of both the non-local and local pseudopotentials of the system provides more precise band structures and wavefunctions [15,16]. 

DFT is a widely used computational method for studying the electronic structure of AGNRs [17,18]. DFT calculations provide a first-principles description of the system by solving the Kohn–Sham equations, taking into account the electron–electron interactions within an exchange–correlation functional. DFT calculations can accurately predict the band structure, electronic density of states, and other electronic properties of AGNRs. However, DFT calculations can be computationally demanding, particularly for larger AGNR systems, and certain approximations within the exchange–correlation functional can introduce errors in the result.

Empirical methods, on the other hand, offer an alternative approach to studying AGNRs [19]. These methods are based on empirical potentials. Empirical potentials capture the essential physics of the system by using simplified parameterized functions that reduce the computational cost compared to DFT. Empirical methods can provide insights into the electronic structure and transport properties of AGNRs with relatively lower computational requirements. They are particularly useful for studying larger systems and phenomena beyond the reach of DFT. For 2D materials, the conventional empirical pseudopotential, which depends on only a few form factors [14,15], does not work well since the contribution from self-consistent charge density and exchange–correlation potential is highly anisotropic for the 2D system. Therefore, the existing empirical potentials do not provide the accuracy desired to provide reliable simulation of the electronic or optical properties of graphene-related devices. Thus, the development of a more accurate SEP is warranted.

Graphene nanoribbons (GNRs) [17,18,19,20,21,22,23,24,25] are narrow strips or ribbons of graphene. They are characterized by their finite width, orientation, and well-defined edges. The width of GNRs can range from a few nanometers to tens of nanometers, and it strongly influences their electronic properties. GNRs can be classified into two main types based on their geometry along the edge: armchair-edge GNRs and zigzag-edge GNRs. In the current work, we solely focus on armchair graphene nanoribbons (AGNRs) to demonstrate the advantage of semi-empirical pseudopotential (SEP). AGNRs have edges formed by rows of carbon atoms that resemble the shape of an armchair. In this configuration, all carbon atoms at the edges are fully bonded, forming a stable structure. AGNRs are known for their semiconducting behavior [20], meaning they can exhibit a band gap in their electronic structure. The width of AGNRs determines the size of the band gap, with narrower ribbons having larger band gaps in general.

In electronics, AGNRs offer tunable bandgaps and unique edge states, making them potential candidates for high-performance transistors and logic devices [26]. AGNRs also exhibit attractive optical properties, with efficient light absorption and emission capabilities, making them suitable for optoelectronic applications such as photodetectors [27] and light-emitting diodes [28]. Furthermore, AGNRs have shown potential in topological physics [29,30,31,32], with the ability to host topological edge states and phenomena such as the quantum spin Hall effect. This opens up possibilities for the development of topological quantum devices and spintronics. Additionally, AGNRs can be tailored to exhibit magnetic properties [33,34] by introducing magnetic dopants or adatoms, enabling applications in spintronics, magnetic sensors, and information storage. AGNRs show promise in thermoelectric applications [35,36,37,38] due to their low thermal conductivity and tunable bandgap, enabling efficient energy conversion. The multifunctionality of armchair graphene nanoribbons makes them a highly versatile platform for advancing electronic, optical, topological, thermoelectric, and magnetic technologies, with ongoing research aimed at uncovering their full potential. 

In this paper, we implement a more sophisticated semi-empirical pseudopotential method (SEPM) within the planar basis to study the atomic and electronic structure of 2D systems, using graphene and armchair graphene nanoribbons as examples. The applications of the currently developed SEP include optical and transport properties of graphene and related structures such as graphene nanoribbons and various junction devices.

## 2. Calculation Methods

### 2.1. B-Splines

To calculate the band structures of 2D materials in the semi-empirical pseudopotential (SEP) model, it is more efficient to adopt a planar basis. The basis consists of localized finite-element functions, i.e., B-splines [39] in the z coordinate multiplied by plane waves in the in-plane coordinates (x and y). B-splines of order κ consist of positive piecewise polynomials of z with degree κ−1. These polynomials vanish everywhere outside the sub-inervals $t_{i}\leq z<t_{i+k}$. The B-spline basis set with order κ defined by a knot sequence $\left\{t_{i}\right\}$ is generated by the following relationship:(1)$$B_{i\kappa }\left(z\right)=\left(\frac{z-t_{i}}{t_{i+\kappa -1}-t_{i}}\right)B_{i,\kappa -1}\left(z\right)+\left(\frac{t_{i+k}-z}{t_{i+k}-t_{i+1}}\right)B_{i+1,\kappa -1}\left(z\right)$$
with $B_{i1}\left(z\right)=\left\{\begin{array}{rr} 1 & \mathrm{if}\;t_{i}\leq z<t_{i+1}\\ 0 & \mathrm{otherwise} \end{array}\right.$, where i=1,2,3… up to the number of knot sequence. 

The first derivative of the B-spline of order κ is given by the following:(2)$$\frac{d}{dz}B_{i\kappa }\left(z\right)=\left(\frac{\kappa -1}{t_{i+\kappa -1}-t_{i}}\right)B_{i,k-1}\left(z\right)-\left(\frac{\kappa -1}{t_{i+\kappa }-t_{i+1}}\right)B_{i+1,\kappa -1}\left(z\right)$$

The derivative of B-splines of order κ is a linear combination of B-splines of order κ−1, which is also a simple polynomial and is continuous across the knot sequence. Using the polynomial expansion
(3)$$B_{i\kappa }\left(z\right)=\sum_{j=1}^{4}\sum_{n=0}^{\kappa -1}D_{n}^{i,j}z^{n}\;\;for\in z\;\left(t_{i+j-1},t_{i+j}\right),$$
we obtain the Fourier transform of $B_{i\kappa }\left(z\right)$ as the following:(4)$$\begin{array}{r} {\tilde{B}}_{i\kappa }\left(g\right)=\frac{1}{\sqrt{L_{c}}}\sum_{j=1}^{4}\sum_{n=0}^{k}D_{n}^{i,j}\int_{t_{i+j-1}}^{t_{i+j}}dze^{ig\left(z-\frac{L_{c}}{2}\right)}z^{n}\\ \equiv \frac{e^{ig{(t}_{i}-L_{c}/2)}}{\sqrt{L_{c}}}\sum_{j=1}^{4}\sum_{n=0}^{\kappa }D_{n}^{i,j}I_{n}^{i,j}\left(g\right). \end{array}$$
where $I_{n}^{i,j}\left(g\right)$ can obtain by the following recursion relationship:(5)$$I_{n}^{i,j}\left(g\right)=\int_{\tau_{i}}^{\tau_{i+j}}dz\;e^{igz}z^{n}={\left(\frac{z^{n}}{ig}e^{igz}\right)|}_{0}^{\tau_{i+j}-\tau_{i}}-\frac{n}{ig}I_{j}^{n-1}\left(g\right)$$
with $I_{0}^{i,j}\left(g\right)={\left(\frac{1}{ig}e^{igz}\right)|}_{0}^{\tau_{i+j}-\tau_{i}}.$ Here, $L_{c}$ is the period length used along *z*, and $I_{n}\left(g\right)$ can be obtained by the recursion relationship. The B-spline functions used for the current calculation are shown in in Figure 1.

### 2.2. Kinetic and Overlap Matrix Elements within Planar Basis

Throughout this paper, we work in atomic units, where energy is in units of Ry, and distance is in units of Bohr. The Hamiltonian of the system in the ultrasoft pseudopotential approach [16] is given by the following:(6)$$\begin{array}{r} \hat{H}=-\nabla^{2}+V_{loc}+{\hat{V}}_{nl} \end{array}$$

Here, $V_{loc}\;and\;{\hat{V}}_{nl}$ denote the local pseudopotential and non-local pseudopotential. Planar basis, used to expand $\phi_{i}$, is defined as the following:(7)$$\begin{array}{r} \langle r|k+G;i\rangle =\frac{1}{\sqrt{A}}e^{i\left(k+G\right)\cdot \rho }B_{i\kappa }\left(z\right) \end{array}$$
where G denotes the in-plane reciprocal lattice vector in 2D, and $\rho =\left(x,y\right)$ is the projection of r in the x-y plane. k is the in-plane wave vector. *A* is the surface area of the sample. The overlap matrix elements between two planar basis functions are given by the following:(8)$$\begin{array}{r} \langle k+G;i|k+G^{\prime };i^{\prime }\rangle =\langle B_{i\kappa }|B_{i^{\prime }}\rangle \delta_{G,G} \end{array},$$
where
(9)$$\langle B_{i\kappa }|B_{i^{\prime }\kappa }\rangle =\int dzB_{i\kappa }\left(z\right)B_{i^{\prime }\kappa }\left(z\right)\equiv O_{ii,}$$
is the overlap integral between two B-spline functions. The kinetic energy matrix elements are given by the following:(10)$$T_{ii^{\prime }}^{}(k+G_{},{k+G}_{}^{\prime })=\langle k+G;i|-\nabla^{2}|k+G;i\prime \rangle =[K_{ii\prime }+O_{ii\prime }{\left|k+G\right|}^{2}]\delta_{G,G},$$
where
(11)$$K_{ii\prime }=\int dz\frac{d}{dz}B_{i\kappa }\left(z\right)\frac{d}{dz}B_{i\prime \kappa }\left(z\right)$$

### 2.3. Implementation of the Semi-Empirical Local Pseudopotential for Graphene

The local pseudopotential of the crystal is given by the following:(12)$$\begin{array}{r} V_{loc}\left(r\right)=V_{ion}\left(r\right) \end{array}+V_{H}\left(r\right)+V_{xc}\left(r\right)$$
where the first term describes the ionic local potential with
(13)$$V_{ion}\left(r\right)=\sum_{\sigma }\sum_{R}V_{L}^{\sigma }\left(r-R-\tau_{\sigma }\right)$$
in which **R** denotes a bulk lattice vector, and $\tau_{\sigma }$ denotes the position of different atoms within the bulk unit cell. We note that $V_{L}^{\sigma }\left(r\right)$ consists of a long-range term that decays like $Z^{\sigma }/r$ for large *r*. For charge-neutral systems, there is a counter long-range term in $V_{H}\left(r\right)$ due to the valence charges, and the sum of $V_{ion}\left(r\right)+V_{H}\left(r\right)$ will be short-ranged. 

For fitting empirical pseudopotentials of bulk materials, one adjusts the Fourier transform of the potential of $V_{loc}\left(r\right)$ (called “form factors” ${\tilde{V}}_{loc}\left(G\right)$) at a small number of reciprocal lattice vectors until the band structure agrees with the experimental data or first-principal calculations [14]. To understand the nature of potential, a set of form factors for the first few shells at a large number of additional reciprocal lattice vectors is needed. A variety of algebraic forms have been used in the past for bulk materials [40,41,42]. We have not found an existing form that has sufficient flexibility to obtain the correct band structure for 2D materials. In our case, we try to mimic the full local potential by an analytic expression of the form.
(14)$$V_{loc}\left(r\right)=\sum_{\sigma ,R}V_{0}\left(r-\tau_{\sigma }-R\right)+\sum_{\sigma ,G\neq 0}{\Delta \tilde{V}}_{loc}\left(z,G\right)e^{iG\cdot (\rho -\tau_{\sigma })}$$
where G denotes an in-plane 2D reciprocal lattice vector. The first term in Equation (14) denotes the main part of the local pseudopotential (denoted $V_{0}),which\;is$ parametrized in terms of three spherical Gaussian functions, with $C_{s}$ and $\alpha_{s}$ being fitted parameters. Namely, $V_{0}\left(r\right)=\sum_{s=1}^{3}C_{s}e^{-\alpha_{s}r^{2}}$. $C_{s}$ and $\alpha_{s}$ are related to the spatial average of $V_{loc}\left(r\right)$ in the 2D plane as a function of z. We can rewrite $V_{0}\left(r\right)$ in terms of 2D plane waves as the following:(15)$$\sum_{R}V_{0}\left(r-\tau_{\sigma }-R\right)=\sum_{s=1}^{3}C_{s}e^{-\alpha_{s}z^{2}}\frac{\pi }{A_{c}\alpha_{s}}\sum_{\sigma ,G}e^{-G^{2}/4\alpha_{s}}e^{iG\cdot (\rho -\tau_{\sigma }).}$$
Let ${\tilde{V}}_{loc}\left(z,G\right)$ denote the Fourier transform of $V_{loc}\left(r\right)$ in the 2D reciprocal space. We have
(16)$${\tilde{V}}_{loc}\left(z,0\right)=\sum_{s=1}^{3}C_{s}e^{-\alpha_{s}z^{2}}\frac{N_{a}\pi }{A_{c}\alpha_{s}}\equiv \sum_{s=1}^{3}D_{s}e^{-\alpha_{s}z^{2}}$$
for the
$\tau_{\sigma }=0$ term at
G=0. Here, $N_{a}$ denotes the number of atoms per unit cell, and $A_{c}$ is the area of the 2D unit cell. We determine the fitting parameters $D_{s}$ and $\alpha_{s}$ by fitting the corresponding ${\tilde{V}}_{loc}\left(z,0\right),$ determined by a DFT calculation within the same mixed basis as defined in (7). The implementation of the DFT package within this basis for graphene and AGNRs was reported in [12]. The DFT results obtained by using the package developed in [12] were checked against results obtained with the VASP package [43], and the calculated results for graphene and related nanoribbons obtained by using both packages are essentially the same. The exchange–correlation functional used is deduced from the Monte Carlo results calculated by Ceperley and Alder [44] and parametrized by Perdew and Zunger [45]. The fitted coefficients for $D_{s}$ can be directly converted to $C_{s}$ through the relationship $C_{s}=D_{s}A_{c}\alpha_{s}/(N_{a}\pi )$. The best-fit parameters for graphene are given in the first row of Table 1, and the quality of the fit for ${\tilde{V}}_{loc}\left(z,0\right)$ is shown in Figure 2a.

The second term of Equation (14) denotes the difference $V_{loc}\left(r\right)-\sum_{\sigma ,R}V_{0}\left(r-\tau_{\sigma }-R\right)$, which is expressed in reciprocal space. Since G=0 is already well fitted by ${\tilde{V}}_{loc}\left(z,0\right),$ we only have to consider the G≠0 contribution. Due to point-group symmetry, the reciprocal lattice vector G can be sorted into many shells with the magnitude of G vectors being the same in each shell, and ${\Delta \tilde{V}}_{loc}\left(z,G\right)$ is the same for all G vectors in the same shell. We found that for G vectors with magnitude G>4 a.u., $\Delta {\tilde{V}}_{loc}\left(z,G\right)$ can be well fitted by a short-range correction function of the form: $${\Delta \tilde{V}}_{S}\left(z,G\right)=f_{S}\left(z\right){\tilde{D}}_{S}\left(G\right)S_{1}(G)$$
with
(17)$$\left\{\begin{array}{l} f_{S}\left(z\right)=\left(1+C_{1}^{S}z^{2}+C_{2}^{S}z^{4}+C_{3}^{S}z^{6}+C_{4}^{S}z^{8}\right)e^{-\alpha_{0}z^{2}}\\ {\tilde{D}}_{S}\left(G\right)=\left(P_{1}^{S}G^{2}+P_{2}^{S}G^{4}+P_{3}^{S}G^{6}\right)e^{-G^{2}/(4\alpha_{2})} \end{array}\right.$$
$S_{1}\left(G\right)=\frac{1}{A_{c}}\sum_{\sigma }e^{-iG\cdot \tau_{\sigma }}$ denotes the structure factor of graphene.

Note that the chosen form of ${\Delta \tilde{V}}_{S}\left(z,G\right)$ goes to zero at G=0, so the fitting does not affect ${\tilde{V}}_{loc}\left(z,0\right).$ We can transform ${\Delta \tilde{V}}_{S}\left(z,G\right)$ back to real space analytically and obtain the following:(18)$${\Delta V}_{S}\left(r\right)=\sum_{\sigma ,G}{\Delta \tilde{V}}_{S}\left(z,G\right)e^{iG\cdot (\rho -\tau_{\sigma })}=f_{S}(z)D_{S}\left(\rho \right),$$
where $D_{S}\left(\rho \right)=\sum_{\sigma ,G\neq 0}{\tilde{D}}_{S}\left(G\right)e^{iG\cdot (\rho -\tau_{\sigma })}.$

The best-fit parameters for short-range correction functions defined above are listed in the first row of Table 2. Finally, for the three shells with 0<G<4 a.u., we need to fit the difference ${\Delta \tilde{V}}_{loc}\left(z,G\right)-{\Delta \tilde{V}}_{S}\left(z,G\right)$ by another long-range correction function of the form ${\Delta \tilde{V}}_{L}\left(z,G\right)=f_{L}(z){\tilde{D}}_{L}(G)$ with the following:(19)$$\left\{\begin{array}{l} f_{L}\left(z\right)=(1+C_{1}^{L}z^{2}+C_{2}^{L}z^{4}+C_{3}^{L}z^{6}+C_{4}^{L}z^{8})e^{-\alpha_{0}z^{2}}\\ {\tilde{D}}_{L}\left(G\right)=(P_{1}^{L}G^{2}+P_{2}^{L}G^{4}+P_{3}^{L}G^{6})e^{-G^{2}/(4\alpha_{2})} \end{array}\right.$$
The best-fit parameters for fitting $f_{L}\left(z\right)$ and ${\tilde{D}}_{L}(G)$ for the three shells are shown in the second row of Table 2. The Fourier transforms of the effective local pseudopotential, i.e., ${\tilde{V}}_{loc}\left(z,G\right)$ for G shells with magnitude $G_{1},G_{2},G_{3}=$ 1.56, 2.702, and 3.12 a.u., respectively, are shown in Figure 2b–d. Similarly, we can transform ${\Delta \tilde{V}}_{L}\left(z,G\right)$ back to real space analytically and obtain the following:(20)$${\Delta V}_{L}\left(r\right)=\sum_{\sigma ,G}{\Delta \tilde{V}}_{L}\left(z,G\right)e^{iG\cdot (\rho -\tau_{\sigma })}=f_{L}(z)D_{L}\left(\rho \right)$$
with $D_{L}\left(\rho \right)=\sum_{\sigma ,G}{\tilde{D}}_{L}\left(G\right)e^{iG\cdot (\rho -\tau_{\sigma })}.$ Then, we obtain the following relationship:$$\;\;\;\;\;\;\;\;\;\;\;\;\;\;\;\;\;\;\;\;\;\;\;\;\;\;\;\;\;\;\;\;\;\;\;\;\;\;\;\;\;\;\;\;{\tilde{D}}_{\gamma }\left(q\right)=\frac{1}{A_{C}}\sum_{\sigma \in A_{c}^{}}\int_{\;}^{\;}d\rho e^{iq\cdot \rho }D_{\gamma }\left(\rho -\tau_{\sigma }\right)/S_{1}(q).$$
where γ=S or L labels the short-range or long-range term.

Combining all above, the total local potential is given by the following:(21)$$V_{loc}\left(r\right)=V_{0}\left(r\right)+{\Delta V}_{S}\left(r\right)+{\Delta V}_{L}\left(r\right).$$

For convenience, $D_{\gamma }\left(\rho \right)$ can be expressed as a polynomial multiplied by a Gaussian function.
(22)$$D_{\gamma }\left(\rho \right)=\sum_{m=0}^{3}b_{m}^{\gamma }\rho^{2m}e^{-\alpha_{2}\rho^{2}}$$
where $b_{m}^{\gamma }$ can be determined by the following relationship:(23)$${\tilde{D}}_{\gamma }\left(G\right)=\frac{1}{A_{C}}\int_{\;}^{\;}dre^{iG\cdot r}D_{\gamma }\left(\rho \right)=\frac{1}{A_{C}}\sum_{m=0}^{3}b_{m}^{\gamma }\int_{\;}^{\;}d\rho e^{iG\cdot \rho }\rho^{2m}e^{-\alpha_{2}\rho^{2}}$$
which leads to the following relationship between $b_{m}^{\gamma }$ and the set of parameters $\left\{P_{1}^{\gamma },P_{2}^{\gamma },P_{3}^{\gamma }\right\}.$
(24)$$\left(\begin{array}{c} b_{0}^{\gamma }\\ b_{1}^{\gamma }\\ \begin{array}{c} b_{2}^{\gamma }\\ b_{3}^{\gamma } \end{array} \end{array}\right)=\frac{A_{C}\alpha_{2}}{\pi }\left(\begin{array}{cccc} 1 & 4\alpha_{2} & 32\alpha_{2}^{2} & 384\alpha_{2}^{3}\\ 0 & -4\alpha_{2}^{2} & -64\alpha_{2}^{3} & -1152\alpha_{2}^{4}\\ 0 & 0 & 16\alpha_{2}^{4} & 576\alpha_{2}^{5}\\ 0 & 0 & 0 & -64\alpha_{2}^{6} \end{array}\right)\left(\begin{array}{c} 0\\ P_{1}^{\gamma }\\ \begin{array}{c} P_{2}^{\gamma }\\ P_{3}^{\gamma } \end{array} \end{array}\right)$$

To check if the real space form of the local pseudopotential $V_{loc}\left(r\right)$ obtained by the current SEPs as given by Equation (21) can truthfully represent the DFT results, we compared the results for $V_{loc}\left(r\right)$ obtained by DFT and by the current method in Figure 3. In Figure 3a, we show the x-dependence of $V_{loc}\left(r\right)$ at z=−0.5Δz and $y=a/(2\sqrt{3})$ (a line passing through the center of a C atom). Here, Δz=W/104 is the grid size for the fast Fourier transform used in the DFT calculation, and W is the width of the domain along the *z*-axis used to define the B-spline basis. 

It is seen that the result obtained by SEP matches the corresponding DFT results very well. In Figure 3b, we show both the x- and y-dependence of $V_{loc}\left(r\right)$ at z≈0 along lines in x- and y-directions, with both passing through the center of the hexagon cell. The agreement with the DFT results is still very good, except there is a small deviation near the center (ρ≈0). We also noticed that the x- and y-dependence almost coincide for $\left|\rho \right|<0.2a$, indicating the potential has nearly cylindrical symmetry near the center of the hexagon cell. We then took the difference of the local potential obtained by DFT and by SEP and plotted the difference function $\Delta V_{b}\left(\rho ,0\right)$ in Figure 3c. It turns out that such a minor correction will still be important for the states derived from the σ-bonds, but it has a negligible effect on states derived from the π-bonds since the π-orbitals have negligible amplitude at the center of the hexagon cell. This minor correction is caused by the fact that all terms included in $V_{loc}\left(r\right)$ in Equation (21) are centered at atomic sites, while the effect of charge redistribution due to valence charges in the solid cannot be fully absorbed by the atom-centered terms. Thus, we should also consider contributions described by functions localized at the center of hexagon cells in the graphene lattice. We shall call such a term the “bond-charge contribution”. 

To determine this contribution, we fit the difference of the local potential obtained by DFT and by SEP, i.e., $\Delta V_{b}\left(\rho ,0\right),$ in Figure 3c by the following analytic function:(25)$$\Delta V_{b}\left(\rho ,0\right)=\sum_{n=0}^{4}C_{n}^{b}\rho^{2n}e^{-\alpha_{b}\rho^{2}}\equiv D_{b}\left(\rho \right).$$
The best-fit parameters {$C_{n}^{b};n=0,\ldots ,4\}$ and $\alpha_{b}$ are listed in Table 3. We approximate the net local potential near the center of the hexagon by a separable form:(26)$$V_{loc}\left(\rho ,z\right)=V_{loc}\left(\rho ,0\right)f_{b}\left(z\right)$$
where $f_{b}\left(z\right)=V_{loc}\left(0,z\right)/V_{loc}\left(0,0\right)$ describes the *z*-dependence of the bond-charge contribution. Here, $f_{b}\left(z\right)$ describes the variation of $V_{loc}\left(\rho ,z\right)/V_{loc}\left(0,0\right)$ along the *z*-axis at ρ=0, i.e., the center of the hexagon cell. By taking the difference between $V_{loc}\left(r\right)$ obtained by DFT and by SEP, we obtain the following:(27)$$\Delta V_{b}\left(\rho ,z\right)\approx D_{b}\left(\rho \right)f_{b}\left(z\right),$$
where we assume that the z-dependent functions obtained by DFT and SEP are the same. 

The net local potential of graphene, $V_{loc}\left(\rho ,z\right),$ evaluated at ρ=0 and obtained by DFT is shown in Figure 4. For convenience, we fit its normalized z-dependent function $f_{b}^{}\left(z\right)=V_{loc}\left(\rho ,z\right)/V_{loc}\left(0,z\right)$ by the following expression:(28)$$f_{b}^{}\left(z\right)=a_{h}e^{-\alpha_{h1}z^{2}}+(1-a_{h})e^{-\alpha_{h2}z^{2}}$$

The best-fit parameters $a_{h},\alpha_{h1},$ and $\alpha_{h2}$ are given in Table 3. Adding this bond-charge contribution, the net local potential in our model becomes the following:(29)$$V_{loc}\left(r\right)=V_{0}\left(r\right)+{\Delta V}_{S}\left(r\right)+{\Delta V}_{L}\left(r\right)+\Delta V_{b}\left(r\right).$$

### 2.4. Fitting of the Non-Local Pseudopotential for Graphene

The semi-empirical pseudopotential contains both local and non-local terms. The non-local pseudopotential contains angular momentum and energy-dependent terms [15]. Incorporating non-local in addition to local terms can provide a more accurate energy range for the valence band edge and best matches to the experimental data. The nonlocal potential is given by a separable form [16]:(30)$${\hat{V}}_{nl}=\sum_{\sigma lm,nn\prime }E_{lm}^{nn\prime }|\beta_{lm}^{n\sigma }\rangle \langle \beta_{lm}^{n\prime \sigma }|$$
where $\beta_{lm}^{n\sigma }\left(r\right)$ denotes the projector functions for an atom at position $\tau_{\sigma }$. For each atom, the best-fit β functions take the following form:(31)$$\beta_{lm}^{n}\left(r\right)=A_{l}^{n}\left(r\right)r^{l}Y_{lm}(\Omega ).$$
for l,m=00,10,11,20,21,22. We fit the beta function for 2S and 2P orbitals for C atom from the Vanderbilt ultrasoft pseudopotential (USPP) [16]. The fitting potential used has the form for $r<R_{s}$, where $R_{s}$ is the cut-off radius.
(32)$$A_{l}^{n}(r)=\left(C_{0}+C_{1}r^{2}+C_{2}r^{4}+C_{3}r^{6}+C_{4}r^{8}\right)e^{-\alpha r^{2}}$$

For the second 2P orbital (2$P_{2}$), we break the β function into two segments with a dividing radius at $R_{s}=0.9228$ (indicated by a vertical dashed line in Figure 5d). We fit the first segment (seg 1) for r<0.9228 by expression (32). For the second segment (seg 2), we fit $A_{l}^{n}(r)$ with the following expression:(33)$$A_{l}^{n}\left(r\right)=\left(C_{0}+C_{1}r_{s}^{2}+C_{2}r_{s}^{4}+C_{3}r_{s}^{6}+C_{4}r_{s}^{8}\right)\;(\mathrm{for}\;0.9228<r<1.2953)$$
where $r_{s}=r-0.9228$. The bet-fit parameter obtained is given in Table 4.

The fitting result for each orbital is shown in Figure 5. The quality of fitting to the input of USPP [16] is excellent. 

### 2.5. Matrix Elements of Local and Nonlocal Pseudopotential

The local pseudopotential $V_{loc}^{}(r)$ is given by Equation (12). It consists of the atomic-like term $V_{0}^{}\left(r\right),$ which is spherical in 3D space, and the correction term, which has a short-range part and a long-range part. We define the following: (34)$$\begin{array}{c} I_{}\left(\alpha_{s}^{};i,,i^{\prime }\right)=\int dzB_{i}\left(z\right)B_{i\prime }\left(z\right)e^{-\alpha_{s}^{}z^{2}}. \end{array}$$

$B_{i}\left(z\right)$ is the B-spline basis functions with the subscript κ dropped for brevity. Within the mixed basis, the matrix elements for the atomic-like term can be written as follows:(35)$$\langle G;B_{i}\left|V_{0}^{}\right|G^{\prime };B_{i\prime }\rangle =\sum_{s}C_{s}^{}I_{}\left(\alpha_{s}^{};i,,i^{\prime }\right)S_{1}(\Delta G){\left(\frac{\pi }{{\alpha_{s}^{}}_{}}\right)}^{}e^{-{\Delta G}^{2}/(4\alpha_{s}^{})}$$
within the B-spline basis. Here, $\Delta G=G^{\prime }-G$ and $C_{s}^{}$ denote the fitting parameters for a given C atom in the unit cell, as given in Table 1.
(36)$$S_{1}\left(\Delta G)=\frac{1}{A_{c}}\sum_{\sigma }e^{i\Delta G\cdot \tau_{\sigma }}=\frac{2}{A_{c}}cos(\Delta G\cdot \tau_{1}\right)$$
denotes the structure factor for graphene. Due to the inversion symmetry of graphene, we choose the origin to be the center between two C atoms in the unit cell, and $\tau_{1}$ is the position of one C atom in the unit cell. Thus, $S_{1}$(ΔG) is real.

Similarly, we obtain the matrix elements for the remaining terms for the local potential as follows:(37)$$\langle G;B_{i}\left|\Delta V_{loc}^{}\right|G^{\prime };B_{i^{\prime }}\rangle =\int dz{B_{i}\left(z\right)\Delta \tilde{V}}_{loc}\left(\Delta G;z\right)B_{i^{\prime }\kappa }\left(z\right)\;\;\;\;=\int dzB_{i}\left(z\right)[{f_{L}\left(z\right){\tilde{D}}_{L}\left(\Delta G\right)S_{1}\left(\Delta G\right)+f}_{S}\left(z\right){\tilde{D}}_{S}\left(\Delta G\right)S_{1}\left(\Delta G\right)+f_{b}\left(z\right){B_{i}\left(z\right)\Delta \tilde{V}}_{b}\left(\Delta G\right)]B_{i^{\prime }}\left(z\right)$$
where
(38)$${\Delta \tilde{V}}_{b}\left(q\right)={\sum_{m=0}^{4}C_{n}^{h}\left(-\frac{\partial }{\partial \alpha_{h}}\right)}^{n}\frac{\pi }{\alpha_{h}}e^{-q_{}^{2}/4\alpha_{h}}$$
denotes the in-plane Fourier transform of the bond-charge contribution, $\Delta V_{b}^{}\left(\rho ,0\right),$ given in Equation (25).

### 2.6. Nonlocal Corrections in Overlap and Potential

The matrix elements for nonlocal potential read as follows [12]:(39)$$\langle K;B_{i}|{\hat{V}}_{nl}|K^{\prime };B_{i^{\prime }}\rangle =\sum_{\sigma lm,nn^{\prime }}E_{lm}^{nn^{\prime }}\langle K;B_{i}|\beta_{lm}^{n0}\rangle \langle \beta_{lm}^{n^{\prime }0}|K^{\prime };B_{i^{\prime }}\rangle e^{i\left(G^{\prime }-G\right)\cdot \tau_{\sigma }}\;=\frac{1}{A_{c}}\sum_{\sigma lm}\sum_{nn\prime }^{}E_{lm}^{nn\prime }P_{lm}^{in}\left(K\right)P_{lm}^{i^{\prime }n^{\prime }*}\left(K\prime \right)e^{i(G^{\prime }-G)\cdot \tau_{\sigma }}.$$

Here, K=k+G; K′=k+G′. $\tau_{\sigma }=\pm \tau_{1}$ for σ=1,2. $P_{lm}^{in}\left(K_{}\right)=i^{l}\sqrt{A_{c}}\langle K_{};B_{i}|\beta_{lm}^{n0}\rangle$ denotes the projection of the β functions (with the center shifted to the origin) into our basis. A detailed description of the evaluation of $P_{lm}^{in}\left(K_{}\right)$ is given in Appendix A. For calculating the band structure, we also need to evaluate the correction to the overlap matrix elements. The correction to the overlap matrix can be written as follows [12]: (40)$$\langle K;B_{i}|\hat{S}|K\prime ;B_{i^{\prime }}\rangle =\frac{1}{A_{c}}\sum_{\sigma lm}\sum_{nn\prime }^{}q_{lm}^{nn\prime }P_{lm}^{in}\left(K\right)P_{lm}^{i^{\prime }n^{\prime }*}\left(K\prime \right)e^{i(G^{\prime }-G)\cdot \tau_{\sigma }}.$$

This expression is identical to (39) except that the energy parameters $E_{lm}^{nn\prime }$ are replaced by the overlap parameters $q_{lm}^{nn\prime }.$ Here, we adopt the same parameters for $E_{l}^{nn\prime }$ and $q_{l}^{nn\prime }$ as in the ab initio input data [12,16] to calculate the matrix elements of non-local pseudopotentials. The input parameter used for $E_{l}^{nn\prime }$ and $q_{l}^{nn\prime }$ are shown in Table 5.

The formulas derived above can be extended to other two-dimensional materials beyond graphene, such as transition-metal dichalcogenides. Due to the inversion symmetry in graphene, all matrix elements become real. Furthermore, there is a mirror symmetry for the *z*-axis. By adopting the B-spine basis along the *z*-axis and taking symmetric and antisymmetric combinations of these basis functions, we can decouple the eigenstates of the Hamiltonian within the symmetric and antisymmetric basis sets for any wave vector in the 2D plane. This significantly improves the speed of computation for the band structures. We found that using a direct solver for diagonalization becomes even faster than the iterative solver based on the conjugate-gradient (CG) approach [46] for this system with both inversion and mirror symmetry. The cut-off used for plane waves used in wavefunctions is 30 a.u. and that in pseudopotential is 180 a.u. 

## 3. Results and Discussions

### 3.1. Band Structure of Graphene

The all-electron (AE) Bloch states of graphene are written as follows:(41)$$\psi_{\nu ,k}\left(\rho ,z\right)=\hat{O}\varphi_{\nu ,k}\left(\rho ,z\right)=\hat{O}\sum_{i,G}Z_{iG}^{\nu ,k}{\tilde{B}}_{i}\left(z\right)\frac{1}{\sqrt{A}}e^{i\left(k+G\right)\cdot \rho },$$
where $\varphi_{n,k}\left(\rho ,z\right)$ denotes the pseudo-wavefunction, ${\tilde{B}}_{i}\left(z\right)$ denotes the orthogonalized B-spline function, and $\hat{O}$ is the overlap operator defined in (A5). The all-electron Kohn–Sham equation [47] reads as follows:(42)$${\hat{H}}_{AE}\psi_{\nu ,k}\left(\rho ,z\right)=E_{\nu }(k)\psi_{\nu ,k}\left(\rho ,z\right)$$
where ${\hat{H}}_{AE}$ is the all-electron Hamiltonian operator, and $E_{\nu }(k)$ denotes the energy of the ν-th band at wavevector k. Substituting Equation (41) into Equation (42) gives rise to the following generalized eigenvalue problem:(43)$$\sum_{i\prime ,G\prime }\langle k+G;{\tilde{B}}_{i}|\hat{H}|k+G\prime ;{\tilde{B}}_{i\prime }\rangle Z_{i\prime G\prime }^{\nu ,k}=E_{\nu }\left(k\right)\sum_{i^{\prime }G^{\prime }}\langle k+G;{\tilde{B}}_{i}|\hat{O}|k+G^{\prime };{\tilde{B}}_{i^{\prime }}\rangle Z_{i^{\prime }G^{\prime }}^{\nu ,k}$$
where $\hat{H}$ is the Hamiltonian operator given in Equation (6).

We apply the current SEP to calculate the band structure of graphene. The real-space structure of graphene can be described by a 2D unit cell with primitive lattice vectors $a_{1}=(1,-\sqrt{3}$)a/2 and $a_{2}=(1,\sqrt{3}$)a/2. The positions of carbon atoms are $(1,-1/\sqrt{3})a/2$ and $(-1,1/\sqrt{3})a/2$. The basis vectors in reciprocal space are $b_{1}=\frac{2\pi }{\sqrt{3}a}\left(\sqrt{3},-1\right),b_{2}=\frac{2\pi }{\sqrt{3}a}(\sqrt{3},1)$. Figure 6 shows the geometry of graphene in real space and reciprocal space.

In Figure 7, the band structure of graphene calculated by SEP with best-fitted parameters deduced in Section 2 is shown and compared with that obtained from a self-consistent calculation based on DFT [12]. The overall band structures obtained by the present method are in close agreement with the ab initio calculation. Since we solve the Bloch states with even and odd parities (with respect to the mirror symmetry about the *z*-axis) separately, the symmetry characteristics of the bands can be easily distinguished. Here, the even (odd) parity states are presented in green (red) color. It is noted that the red curves (with odd parity) are related to the $p_{z}$ orbitals of the carbon atoms, and they form π-bonded bands, whereas the green curves (with even parity) are related to the s,
$p_{x},{andp}_{y}$ orbitals.

As shown in Figure 7, the agreement between the SEP and DFT results obtained by using the method described in [12] for the lowest 20 bands is excellent. The advantage of the SEP is that it is easy to use and requires no self-consistent calculation to establish the charge density. The time needed to calculate the band structures via a direct solver on a laptop PC is less than 6 s, which is only a fraction (~1/5) of the time needed for the CG calculation used in the DFT package (after the self-consistent density is established). The saving is mainly due to the use of the inversion and mirror symmetry properties of graphene.

### 3.2. Band Structure of Armchair Graphene Nanoribbon

We consider a graphene nanoribbon with armchair edges with a width of Na, where a is the lattice constant of graphene. The supercell of the armchair graphene nanoribbon (AGNRs) contains 4N+2 atoms, as illustrated in Figure 8a. To adopt the 2D plane-wave basis, we introduce a vacuum region with the width of (M−N)a inside the supercell such that the dimension of the rectangular supercell is $\sqrt{3}a\times Ma$. The reciprocal lattice vectors of a superlattice consisting of a 1D periodic of AGNRs along the x-axis can be written as $g_{n_{1}n_{2}}=n_{1}{\tilde{b}}_{1}+n_{2}{\tilde{b}}_{2}$, where ${\tilde{b}}_{1}=\frac{2\pi }{Ma}\hat{x}$ and ${\tilde{b}}_{2}=\frac{\pi }{a}\hat{y}$ are the basis vectors in reciprocal space, and $n_{1}$ and $n_{2}$ are arbitrary integers. Let $g_{i}$
(i=1,…,2M) denote those non-equivalent g vectors falling within the first Brillouin zone of graphene. (See Figure 8b.) Then, all g vectors of the AGNRs superlattice can be expressed as $g_{i}+G$, where G denotes the reciprocal lattice vectors of graphene. 

The Bloch states of the AGNRs superlattice can be written as linear combinations of basis functions that are products of 2D plane waves (labeled by wave vectors ${k+g}_{j}+G)$ and B-splines in z (i.e., the mixed-basis introduced in [11]). For wide AGNRs, this basis set can be quite large. The band structure of the AGNRs is expected to be quite close to the zone-folded band structure of graphene, with deviation mainly coming from the quantum confinement effect from the vacuum region and the effect due to the creation of edges. To describe such a change, it is computationally more efficient to start with the zone-folded band structures of graphene that can be calculated by using the graphene SEP code developed above at wave vectors of ${k+g}_{j}$
$\left(j=1,\ldots ,2M\right),$ where k is within the first mini zone of the AGNR superlattice. A similar approach was applied to the Si 7 × 7 surface, and it was demonstrated that such a method works very well for large superstructures [48]. If the vacuum region introduced in the superlattice is thick enough, the coupling between adjacent AGNRs can be negligible, and k of interest will be along the long axis of the AGNR (taken to be parallel to $\hat{y}$ here). The low-lying pseudo-Bloch states of graphene (with band labeled by ν) obtained at ${k+g}_{j}$ (denoted as $\varphi_{\nu ,{k+g}_{j}}$) can then be used as a set of contracted basis functions for calculating the band structures of the AGNR. This set of contracted bases is a nearly complete basis for AGNR if we use a large number of graphene bands until the convergence is established for the calculated results. 

Thus, we can write the Hamiltonian of the AGNR as $H=-\nabla^{2}+U_{1}+{\hat{U}}_{2}$, where
(44)$$U_{1}\left(r\right)=\left[V_{0}^{\prime }\left(r\right)+{\Delta V}_{S}^{\prime }\left(r\right)+{\Delta V}_{L}^{\prime }\left(r\right)+\Delta V_{b}^{\prime }\left(r\right)\right]$$
denotes the sum of local potentials of C atoms defined in Equation (12) for atomic sites (indexed by σ) in the nanoribbon region $A_{NR}^{}$ [with $\left|x\right|\leq W/2]$ inside the supercell used for AGNRs. ${\hat{U}}_{2}$ denotes the nonlocal pseudopotential of the nanoribbon. The AGNRs Hamiltonian matrix within the contracted basis is then given by the following:(45)$$\langle \varphi_{\nu ,{k+g}_{j}}|{\hat{H}}_{GNR}|\varphi_{\nu^{\prime },{k+g}_{j^{\prime }}}\rangle =\langle \varphi_{\nu ,{k+g}_{j}}|-\nabla^{2}|\varphi_{\nu^{\prime },{k+g}_{j^{\prime }}}\rangle +\langle \varphi_{\nu ,{k+g}_{j}}|U_{1}+{\hat{U}}_{2}|\varphi_{{}^{\prime },{k+g}_{j^{\prime }}}\rangle$$
where the first term describes the kinetic energy, and the second term consists of the semi-empirical local ($U_{1})$ and nonlocal (${\hat{U}}_{2})$ pseudopotentials for AGNR. The detailed expressions for these matrix elements are given in Appendix B.

In addition to the z-mirror symmetry, the AGNR also has x-mirror symmetry for any wave vector, i.e., k, along the *y*-axis. Thus, we can define symmetrized basis states with respect to the x-mirror as follows: (46)$${|\varphi }_{\nu ,{k+g}_{j}}^{s,\pm }\rangle =f_{j}\left({|\varphi }_{\nu ,{k+g}_{j}}^{s}\rangle \pm {|\bar{\varphi }}_{\nu ,{k+\bar{g}}_{j}}^{s}\rangle \right)/\sqrt{2}\;\mathrm{for}\;g_{jx}\geq 0,$$
where ${\bar{g}}_{j}=\left({-g}_{jx},g_{jy}\right),$
${\bar{\varphi }}_{\nu ,{k+\bar{g}}_{j}}^{s}\left(x,y,z\right)=\varphi_{\nu ,{k+g}_{j}}^{s}\left(-x,y,z\right)$ (corresponding to flipping the sign of $G_{x}$ in the plane-wave basis). $f_{j}=1/\sqrt{2}$ for $g_{jx}=0,$ and $f_{j}=$ 1 otherwise. The symbol “±” denotes even/odd parity states with respect to the x-mirror, while the superscript s=e,o is the label for the even/odd parity states with respect to the z-mirror. The Hamiltonian matrix elements between symmetrized states of the same parity read as follows:(47)$$\langle \varphi_{\nu ,{k+g}_{j}}^{s,\pm }|{\hat{H}}_{GNR}|\varphi_{\nu \prime ,{k+g}_{j\prime }}^{s,\pm }\rangle =f_{j}f_{j\prime }\left[\langle \varphi_{\nu ,{k+g}_{j}}^{s}|{\hat{H}}_{GNR}|\varphi_{\nu \prime ,{k+g}_{j\prime }}^{s}\rangle \pm \langle \varphi_{\nu ,{k+g}_{j}}^{s}|{\hat{H}}_{GNR}|\varphi_{\nu \prime ,{k+\bar{g}}_{j\prime }}^{s}\rangle \right].$$

The overlap matrix is given by the same expression above, with ${\hat{H}}_{GNR}$ replaced by ${\hat{O}}_{GNR}$.

Here, we divide the basis set into four different subsets according to different symmetry types labeled by {$\pi_{+},\pi_{-},\sigma_{+},\sigma_{-}\}.$
$\pi_{+}$ and $\pi_{-}$ states represent π-bonded ($p_{z}$-like) states with even and odd parity, respectively, under the x-mirror operation. $\sigma_{+}$ and $\sigma_{-}$ states represent σ-bonded ($p_{x},p_{y},$ or *s*-like) states with even and odd parity, respectively, under the x-mirror operation. Eigenstates of ${\hat{H}}_{GNR}$ with different symmetry types are decoupled based on group theory. Thus, we can block-diagonalize the Hamiltonian matrix into four diagonal sub-blocks, and each diagonal sub-block can be diagonalized separately. By taking advantage of these symmetry properties, the electronic structures of graphene nanoribbons can be solved very efficiently with the present SEP via a direct diagonalization method. Furthermore, the symmetry characteristics of different bands can be easily identified in the band structure by using different colors. This can help sort out the complex characteristics in AGNR electronic states and improve the understanding of the electronic properties of AGNR.

The graphene bulk basis adopted above can describe the band folding due to the quantum confinement effect in the nanoribbon well, but it requires a large number of basis states to fully capture the effect of localized edge states. The AGNRs band structures obtained by solving Equation (43) are shown in Figure 9a,b to compare with the DFT results. The number of graphene bands included in the calculation is 10 for the $\pi_{+}(\pi_{-}$) states and 60 for the $\sigma_{+}(\sigma_{-})$ states. Adding more bands produces no significant effect. For the odd states, the change of energy on the AGNR band structure is less than 0.005 eV within the energy window of interest. For the σ-bonded (even in z) states, most minibands remain nearly unchanged when the number of graphene bands included varies from 14 to 60, while the two pairs of minibands closest to the band-gap region change significantly and still do not reach the desired convergence level even with 60 graphene bands included in the basis. These edge-induced states correspond to the bonding and antibonding states of the dimers on two edges. Since there will be an edge-induced correction in the local potential, it is unnecessary to spend a significant effort to obtain fully convergent results for the preliminary investigation in this section. 

We also performed DFT calculations of the same AGNR within the basis constructed by 2D plane waves multiplied by B-spline functions of z, as described in [12]. The edge atoms are allowed to relax to minimize the total energy of the AGNR. In the relaxed geometry, the displacement of the edge atom in the left and upper corner of the AGNR supercell (the green box) as shown in Figure 8a is described by $\Delta \tau =\left(\Delta x,\Delta y\right),$ with Δx=0.2107 a.u. and Δy=0.1788 a.u. The displacement of three other edge atoms can be deduced by applying the x-mirror and y-mirror symmetry. The resulting band structures are shown in Figure 9c. To make sure that the relaxation of only the edge atoms is sufficient, we also performed the DFT calculation of the AGNR, in which the outermost four rows of atoms are allowed to relax, and we found that the major displacement occurs on the edge atoms, while the other inner rows only relax slightly, and the resulting band structure is quite similar to the one with relaxation only on the edge atoms. For comparison, we shifted the DFT band structure rigidly in energy, so the valence-band maximum is aligned with our SEP calculation. 

We found that, as shown in Figure 9a, the band structure of the unrelaxed AGNR looks qualitatively similar to the corresponding DFT results (with edge relaxation) in Figure 9c. However, the band gap obtained is about 0.45 eV, which is substantially smaller than the DFT result of ~0.6 eV. Furthermore, we found a pair of mid-gap bands (in green) that are related to the σ-bonding states of the dimer atoms on the edge. There is also a pair of σ-antibonding states of the edge dimers with energy near 0 eV at the Γ point (not shown). These edge-dimer-related minibands are nearly doubly degenerate, corresponding to dimer states located at the left and right edges of the AGNR. Since the width of the ribbon is much larger than the spread of the wavefunctions localized on edges on opposite sides, these states form a closely spaced pair of minibands. In the DFT band structure, the corresponding σ-bonding states are lowered by ~2.5 eV at the Γ point, and there are no mid-gap states left. Since the DFT calculation did not separate the even and odd states, some of them even states look artificially entangled with the odd states.

The SEP band structures of the AGNR with the same relaxed atomic positions as determined by DFT but with the pseudopotential (PP) given by Equation (29) are shown in Figure 9b. With the relaxation, the band gap becomes enlarged to ~0.5 eV, and the overall band structure agrees better with the DFT results. The pair of σ-bonding states of edge-dimers (in green) is also lowered but only by ~0.5 eV at Γ point, and there are no states left in the mid gap. We note that for device applications, the π-bonded states (near the band gap region) play a dominant role. The band structure for these π-bonded states is not significantly changed due to the relaxation of atomic positions at the edges. However, there are still noticeable differences between the band structures obtained by SEP and by DFT, which are mainly caused by the deviation of local potential near the edge atoms as the graphene is cleaved to form a nanoribbon. We discuss the consequence of this effect and its remedy below.

### 3.3. Modification of Pseudopotential for Edge Atoms of Armchair Graphene Nanoribbon

Figure 10 shows the contour plot of the local potential of the 9 × 2 AGNR with edge relaxation. The ~5% shift of atomic positions near the edges is noticeable in this plot. To examine the deviation of local potential ($V_{loc}$) calculated by SEP from the DFT results, we plot the line cuts of $V_{loc}(x,y,z)$ as functions of x at the four different values of y (y=−L/2,−3L/17,0, and 6L/17) and z≈0 in Figure 11. These lines are indicated by black lines in Figure 10. Here, $L=\sqrt{3}a$ is the length of the AGNR supercell along the *y*-axis. 

As shown in Figure 11, we found that the local potentials obtained by SEP agree very well with the DFT results in the interior region of the AGNR, with noticeable deviation from the DFT results only near the two edges (with |x|>4.5a). It implies that the SEP potential centered at an atom follows the position of the relaxed atom very well, while the cleavage of graphene to form an AGNR causes some charge redistribution that can lead to the change of the local potential near the edges.

To investigate the effect of local potential induced by the edge creation, we took the difference between the DFT (red curve) and SEP results (green curve) in Figure 11 for x between 3a and 6a at y=−L/2, −3L/17, 0, and 6L/17. The results for the differences at the selected values of y are shown as dotted lines in Figure 12. These difference curves (for one of the two edges) can be reasonably fitted by the following functional form:(48)$$\Delta V_{e}\left(\rho ,0\right)\approx \left[S_{a}e^{-\alpha_{a}{\left(\rho -\rho_{a}\right)}^{2}}+{S_{b}e}^{-\alpha_{a}{\left(\rho -\rho_{b}\right)}^{2}}+{S_{c}e}^{-\alpha_{a}{\left(\rho -\rho_{c}\right)}^{2}}+S_{d}e^{-\alpha_{b}{\left(\rho -\rho_{d}\right)}^{2}}\right],$$
where $\rho_{a}=(x_{a},-L/2),$
$\rho_{b}=(x_{b},-3L/17),$
$\rho_{c}=\left(x_{b},0\right),$ and $\rho_{d}=(x_{b},y_{d})$ denote the four locations of effective bond charges near the AGNR right edge. $y_{d}=y_{e}+\Delta y\approx 6L/17$ denotes the y-coordinate of the relaxation edge atom, including the relaxation Δy. The AGNR potential also has a y-mirror symmetry, so we duplicate the terms at $y_{b}$ and $y_{d}$ and add the same potentials at $-y_{b}$ and $-y_{d}$. We note that the potentials at $y_{a}=-L/2$ (equivalent to $y_{a}=L/2$ due to periodicity) and $y_{c}=0$ will be mapped to themselves by the y-mirror. Here, $x_{a}=5a$ denotes the bond-charge location at y=−L/2, and $x_{b}=4.65a$ denotes the common bond-charge location near the edges for other y values. The positions of these bond charges are related to the minimum of the difference in the local potentials shown in Figure 12. $S_{a}$, $S_{b}$, $S_{c}$, and $S_{d},$ describing the variation of potential strength along the y-axis. Here, we choose $\alpha_{a}=0.4$ and $\alpha_{b}=1.1$, which approximately describe the width of the potential wells shown in Figure 12. The best-fit results for $\Delta V_{e}\left(\rho ,0\right)$ are shown as green curves in Figure 12, with best-fit values of $S_{a}$, $S_{b}$, $S_{c},$ and $S_{d}$ listed in Table 6.

We approximate the net three-dimensional local potential, including the bond-charge redistribution near the edges by a separable form:(49)$$V_{loc}\left(\rho ,z\right)=V_{loc}\left(\rho ,0\right)f_{e}\left(z\right)$$
where $f_{e}\left(z\right)=V_{loc}\left(0,z\right)/V_{loc}\left(0,0\right)$ describes the *z*-dependence of the bond-charge contribution near the edges. $f_{e}^{}\left(z\right)$ can be extracted from the net local potential of AGNR, $V_{loc}\left(x_{b},0,z\right),$ obtained by DFT with $x=x_{b}$ and y=0. The result is shown in Figure 13, and it can be well fitted by the sum of two Gaussian functions. The normalized z-dependence of the local potential is given by the following:(50)$$f_{e}^{}\left(z\right)=V_{loc}\left(x_{b},0,z\right)/V_{loc}\left(x_{b},0,0\right)=C_{1}^{e}e^{-\alpha_{e1}z^{2}}+C_{2}^{e}e^{-\alpha_{e2}z^{2}}$$
where $C_{2}^{e}=1-C_{1}^{e}$ due to the normalization requirement similar to Equation (28). We obtain $C_{1}^{e}=0.979$, $\alpha_{e1}=1.0926,$ and $\alpha_{e2}=0.1026$. Thus, the correction to local pseudopotential due to the bond-charge redistribution near the edges can be approximately given by the following:(51)$$\Delta V_{e}\left(\rho ,z\right)\approx \Delta V_{e}\left(\rho ,0\right)f_{e}(z)$$
where $\Delta V_{e}\left(\rho ,0\right)$ is given in Equation (48) and $f_{e}(z)$ in Equation (50).

Finally, we add the edge-induced correction in the local pseudopotential, $\Delta V_{e}\left(\rho ,z\right),$ to the Hamiltonian of the AGNR in our semi-empirical pseudopotential model. The matrix elements of $\Delta V_{e}\left(\rho ,z\right)$ within the contracted basis functions derived from the selected graphene eigenstates at special reciprocal lattice vectors for AGNR ($g_{j}$) are given by the following:(52)$$\langle \varphi_{\nu ,{k+g}_{j}}|\Delta V_{e}|\varphi_{\nu^{\prime },{k+g}_{j^{\prime }}}\rangle =\sum_{iG,i^{\prime }G^{\prime }}Z_{iG}^{\nu ,{k+g}_{j}\;}Z_{i^{\prime }G}^{\nu^{\prime },{k+g}_{j}}\sum_{m=1,2}^{\;}C_{m}^{e}I_{}\left({\alpha_{em}}_{\;}^{};i,,i^{\prime }\right){\tilde{v}}_{e}(\Delta g_{jj^{\prime }}+\Delta G)$$
where $I_{}\left(\alpha_{}^{};i,,i^{\prime }\right)$ is defined in Equation (34), $\Delta g_{jj^{\prime }}+\Delta G=g_{j^{\prime }}-g_{j^{}}+G^{\prime }-G,$ and
(53)$${\tilde{v}}_{e}\left(q\right)=\frac{1}{A_{SC}}\int_{\;}^{\;}dre^{iq\cdot \rho }\Delta V_{e}\left(\rho ,0\right)+c.c.$$

Here, $\Delta V_{e}\left(\rho ,0\right)$ is the edge-induced correction in pseudopotential for the right edge, as given in Equation (48), and c.c. denotes the term contributed from the left edge, which is the complex conjugate of the previous term due to the inversion symmetry in the 2D plane of the AGNR. With the use of the analytic fitting functions introduced above, ${\tilde{v}}_{e}\left(q\right)$ can be written as follows:(54)$$\begin{array}{r} {\tilde{v}}_{e}\left(q\right)=\frac{2}{A_{SC}}\{S_{a}I_{0}\left(\alpha_{a},q\right)\cos\left(q_{x}x_{a}\right)\cos\left(q_{x}y_{a}\right)\\ +\left[{2S}_{b}I_{0}\left(\alpha_{a},q\right)\cos\left(q_{x}y_{b}\right)+S_{c}I_{0}\left(\alpha_{a},q\right)+2S_{d}I_{0}\left(\alpha_{b},q\right)\cos\left(q_{x}y_{d}\right)\right]cos(q_{x}x_{b})\} \end{array}$$
with $I_{0}\left(\alpha ,q\right)=\frac{\pi }{\alpha }e^{-q_{\;}^{2}/4\alpha }$. Due to the x-mirror symmetry, there are four corresponding bond charges at the left edge. 

After adding the correction term $\Delta V_{e}\left(\rho ,z\right)$, the calculated band structure for the AGNR is shown in Figure 9d. To improve the convergence for the edge-dimer-related states, we included 90 graphene bands in our basis for the calculation of even states, while only 10 graphene bands are needed to achieve the desired convergence for odd states. We found that by adding more graphene bands to the calculation, the results remain nearly the same. Comparing with Figure 9c, where we did not include the edge-induced correction in pseudopotential, $\Delta V_{e}\left(\rho ,z\right)$, we found that the main effect of $\Delta V_{e}\left(\rho ,z\right)$ is to give a significant improvement of the miniband structures for the π-bonded states (in red or blue) of the AGNR, which agree very well with the DFT results as shown in Figure 9c, and the band gap obtained by SEP (~0.7 eV) also matches the DFT result well. The other important feature is the lowering of the σ-bonding states of edge dimers at the zone center from −4.3 eV in Figure 9b to −6.3 eV in Figure 9d, while the σ-antibonding states of edge dimers are also lowered by about 2 eV, and they appear in Figure 9d at −2.3 eV at the Γ point. The energy spacing between these edge-dimer bonding and anti-bonding states is about 4 eV at k = 0 (the zone center), and it reduces to about 3.3 eV at k=π/L (the zone boundary). This feature also agrees reasonably well with the DFT result shown in Figure 9b. The energy position of these edge-dimer bonding states (near −6.75 eV at the zone boundary) is close to the DFT result (−6.8 eV), while the edge-dimer antibonding states sit around −3.46 eV, which is higher than the corresponding DFT results by about 0.3 eV. This discrepancy is likely due to other corrections in the pseudopotential that are not yet included by the current SEP. On the other hand, the band structure of other even states (in green) not related to the edge-dimer states are in very good agreement with the DFT results. Since the edge-dimer antibonding states have a minimum at the zone boundary, and their energies are higher than the conduction band minimum at the zone center, they may not play a significant role in certain device applications. In case they are needed, a “scissor operation” can be applied to shift these minibands rigidly to match the DFT results, and phenomena such as the Gunn effect and negative differential resistance [47,48] can be simulated with this simple approach, and it remains a reasonable approximation to use the wavefunctions obtained by SEP with the current set of basis to calculate the carrier-scattering process in the nano Gunn-diode application by using AGNRs. Since these edge-dimer states are very sensitive to the chemical modification of the AGNR edges, our model can be used to simulate the effect of edge modification on the I-V characteristics by imposing a model potential on the edge atoms. 

### 3.4. Comparison with Experiments

Many experimental studies on graphene and AGNR-related devices have been reported in the last decade [49,50,51,52,53,54,55,56,57,58,59,60,61], which validate and complement the theoretical predictions, providing critical empirical evidence. Techniques such as scanning tunneling microscopy (STM) [51,52,53,54,55,56,57], electrical transport measurements [58,59,60], and angle-resolved photo emission spectroscopy (ARPES) [61] are commonly used in AGNR research. Experimental measurements verify the existence of energy bandgaps, electronic states, and other electronic and transport phenomena predicted by theory. The band gap of AGNR is determined by the energy difference between the conduction band and the highest valence band at the Γ point. We assume that the edge relaxation and the edge-induced correction in pseudopotential remain unchanged for AGNRs with N=3~12. The corresponding numbers of dimer lines in these AGNRs are $N_{d}=2N+1=7\sim 25$. The calculated band energies for the highest occupied molecular orbital (HOMO) and lowest unoccupied molecular orbital (LUMO) for AGNRs as functions of $N_{d}$ are shown in Figure 14. Here, we only consider the cases with odd $N_{d}$ so that the x-mirror symmetry holds. The computation time is less than 10 s for each case shown.

According to simple models that impose rigid boundary conditions on the edges of AGNRs, the band gap becomes zero when $N_{d}=3m+2$, where m is a positive integer [20,62]. Our calculations show that the band gaps indeed shrink to relatively small values at $N_{d}=11,\;17,\;23.$ However, at $N_{d}=11$ and 17, the band gap is still appreciable (~0.14 eV and 0.13 eV). This is because the boundary conditions become not so rigid in the realistic situation. Our calculated value of ~0.14 eV is consistent with the small band gaps (~0.18 eV) observed in AGNRs of similar dimensions [51]. The band gap obtained by our calculation for the $N_{d}=7$ case is 1.43 eV, which is close to the result of 1.47 eV obtained by a previous DFT calculation [63] and also consistent with the experimental result of ~1.4 eV based on Fourier-transformed scanning tunneling spectroscopy [52]. However, our result is ~0.9 eV, lower than that obtained by the dI/dV measurements for the 7-AGNR, which shows a band gap ~2.3 eV [52]. Since our calculation does not include the many-body effect for quasiparticle excitation (nor does the DFT calculation without GW correction), our result is expected to be significantly lower than the DFT-GW calculation [54], which predicts a band gap of ~3.94 eV when the many-body effect is included via the GW approximation [7]. However, most dI/dV measurements were taken for AGNR samples placed on Au substrate [51,52,53,54,55,56,57], and there is a strong screening effect on the many-body effect, which explains the large difference in band gap predicted by DFT-GW calculation and experimental results [63]. 

For the case of 9-AGNR, the band gap obtained by the current SEPM is 0.68 eV, which is 0.7 eV lower than the 9-AGNR band gap measured by dI/dV measurements (~1.4 eV) [54] due to the many-body effect. For the 13-AGNR, our calculation predicts a band gap of 0.9 eV, and the LUMO level is ~0.8 eV lower than the 7-AGNR (See Figure 14). This energy difference in LUMO levels between 7- and 13-AGNRs is close to the result (~0.7 eV) obtained by the dI/dV measurements on a 7–13 AGNR heterojunction [22]. Since we adopted a B-spline basis along the *z*-axis, our calculated band energies are absolute values with respect to the vacuum level. Therefore, it is meaningful to make such a comparison. For the case of 15-AGNR, the band gap obtained by the current SEPM is 0.49 eV, which is ~0.5 eV lower than the dI/dV measurements of 1.03 eV [56]. For the case of 21-AGNR, the SEPM band gap is 0.32 eV, which is ~0.4 eV lower than the value of 0.7 eV obtained by dI/dV measurements [57]. Therefore, in general, the band gaps for AGNRs predicted by the current SEPM are fairly close to those obtained by other DFT calculations (without including the GW correction), and the values are consistently lower than the dI/dV measurements for AGNRs on Au substrate by an amount that varies from 0.4 eV at $N_{d}=21$ to 0.9 eV at $N_{d}=7$. Including the GW correction tends to obtain much larger band gaps in comparison to the experimental results [52], which can be attributed to the screening effect from the metal substrate on the GW correction. Thus, it is necessary to carry out DFT-GW calculations for AGNRs on Au substrate, as reported in [63], in order to determine the amount of band-gap correction due to the many-body effect. 

## 4. Conclusions

We developed a semi-empirical pseudopotential (SEP) method that is easy to implement and capable of obtaining accurate band structures for graphene and armchair-edged graphene nanoribbons. For the π-bonded states (with odd symmetry with respect to the *z*-axis mirror), our SEP method can nearly reproduce all salient features of the DFT results with good accuracy. The time needed to compute the whole band structure associated with π-bonded states of graphene is only about a few seconds on a personal computer (PC), and it takes around 100 s to compute the whole band structure for π-bonded states of an AGNR with supercell size of $16a\times \sqrt{3}a$. Thus, it will be a highly efficient method for modeling AGNR-related devices when the π-bonded states play the primary role. For the modeling of AGNR devices that involve the indirect valley minimum at the zone boundary, the edge-dimer antibonding states (which possess even symmetry with respect to the z-axis mirror) will be needed. For such a case, the current method becomes less efficient. It will take about one hour on a PC to obtain the nearly correct energy and dispersion of the localized σ-bonding and σ-antibonding states of the edge dimmers of the AGNR, while the corresponding DFT calculation will take more than one day. 

The SEP method with the same edge-induced correction potential given in Equation (51) is also applicable for AGNRs of sizes different from the example used here since the dimer formation on AGNR edges should not be significantly affected by the width of the interior region. Thus, the current model is suitable for modeling any size of AGNR (as long as the width is at least 3a) for device applications. 

The current approach can still be improved by adding localized basis functions at the edges to make the convergence of σ-bonding and antibonding states at AGNR edges much faster. Furthermore, an SEP for the other popular graphene nanoribbons with zigzag edges will also be useful. The current approach can also be extended to develop similar SEPs for transition-metal dichalcogenides (TMDs) and related moiré superlattices. All these improvements are worthy topics for future research. For application to TMDs, some complications will occur due to the more complicated structure factors, which cannot be made real. Therefore, finding semi-empirical potentials to fit both the real and imaginary parts of the complex quasi-2D form factors, ${\tilde{V}}_{loc}\left(z,G\right),$ for two kinds of atoms will require more tedious procedures, although it can still be done. Once it is done, the more interesting application is to develop an SEPM for TMD moiré superlattices considering their high scientific impact. Here, we have demonstrated the success of using graphene eigenstates at different $g_{j}$ points enclosed in the graphene Brillouin zone (BZ) (see Figure 8b) as a contracted basis set to calculate the AGNR band structures efficiently. The same idea can be applied to deal with twisted bilayer TMDs with a large supercell. Since there are no edge states to deal with in the moiré superlattices, we expect that only a small number of TMD bands need to be included in the contracted basis set. By using this contracted basis, the miniband structures of TMD moiré superlattices can be calculated very efficiently with good accuracy (similar to the π-bonded states in AGNR). The wavefunctions of AGNRs calculated by SEPM are very close to the DFT results. Therefore, they can be used to calculate the electron–phonon scattering with good accuracy. Furthermore, since all our AGNR wavefunctions (including edge states) are written as linear combinations of graphene Bloch states, we can relate the electron–phonon scattering matrix elements to those for graphene, and it will be convenient to model the transport properties in AGNR-related optoelectronic devices. The software developed here will be valuable for IC designs of 2D material-based nanoelectronics devices that will be of interest to the semiconductor industry. 

The close interplay between theoretical and experimental studies fosters a deeper understanding of AGNRs. Moreover, experimental data provide feedback to refine and improve theoretical methods. The collaboration between theorists and experimentalists allows for the identification of new phenomena and the validation of theoretical models. Together, theoretical and experimental studies offer a comprehensive perspective on AGNRs’ properties and pave the way for potential applications in nanoelectronics and beyond.

## Figures and Tables

**Figure 1 nanomaterials-13-02066-f001:**
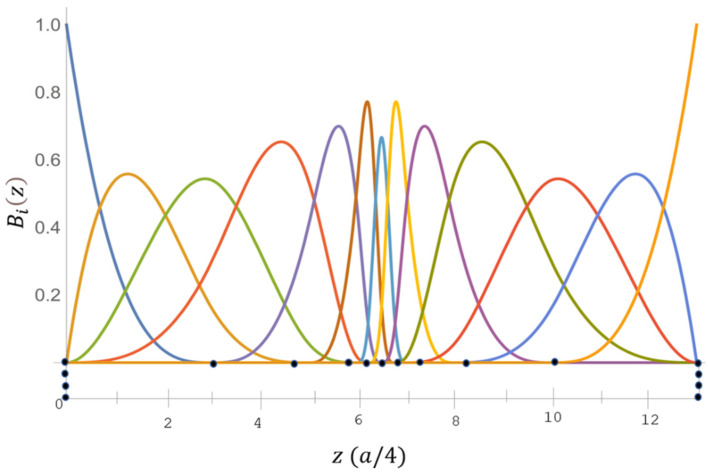
B-spline consists of an exponential-type of a 17-point knot sequence of order κ=4. Here, (●) denotes the knot points in the sequence.

**Figure 2 nanomaterials-13-02066-f002:**
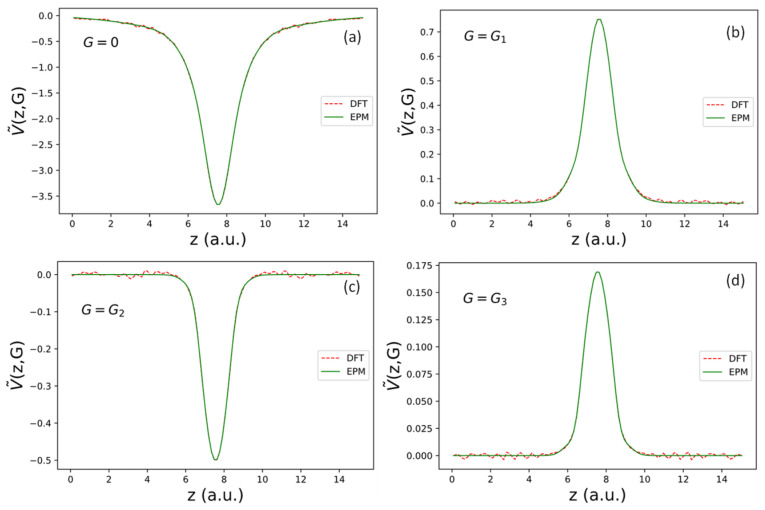
Fourier transform of effective local pseudopotential ${\tilde{V}}_{loc}(z,G)$ for reciprocal lattice vectors (G ) in the first few shells. The magnitudes of G in subfigures are: (**a**) $G_{}=0,$ (**b**) $G_{}=G_{1}=$ 1.56 a.u., (**c**) $G_{\;}=G_{2}=$ 2.70 a.u., and (**d**) $G_{}=G_{3}=$ 3.12 a.u. for shells 0, 1, 2, and 3, respectively.

**Figure 3 nanomaterials-13-02066-f003:**
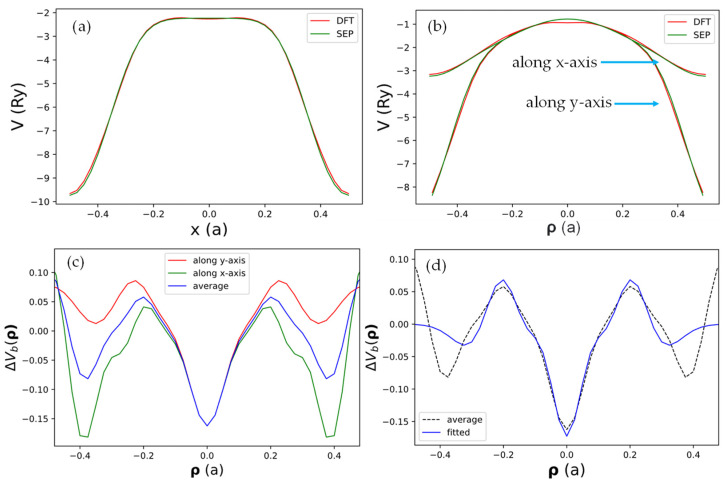
The net local pseudopotential in real space $V_{loc}\left(\rho ,z\right)$ along various lines in the plane with z=−0.5Δz (half a grid from the center of the supercell). (**a**) Along the line at $y=\frac{a}{2\sqrt{3}},$ which passes through a row of C atoms in the graphene sheet. (**b**) Along two perpendicular lines (along the *x*-axis and *y*-axis) both passing through the origin, which is the center of the hexagon cell. (**c**) The difference of $V_{loc}\left(\rho ,z\right)$ obtained by DFT and SEP plotted along the *x*-axis (green) or *y*-axis (red). The blue curve indicates the average of the red and green curves. (**d**) The best-fit results of the average $\Delta V_{b}\left(\rho ,z\right)$.

**Figure 4 nanomaterials-13-02066-f004:**
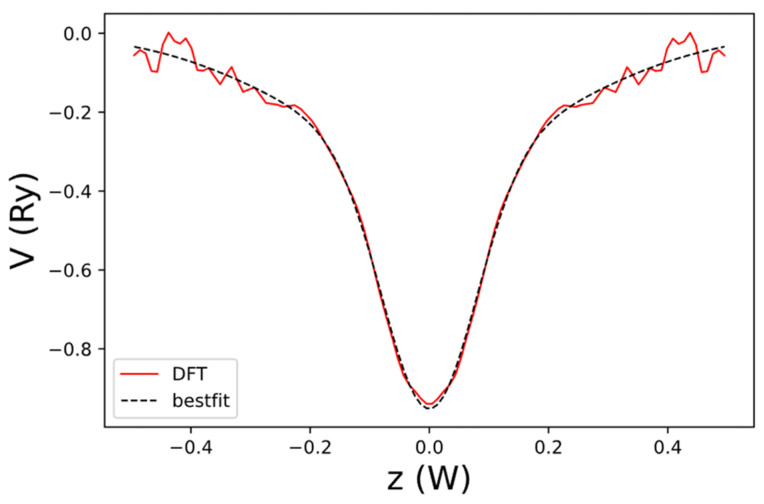
The net local potential of graphene, $V_{loc}\left(0,z\right),$ as a function of z obtained by DFT (red curve) and the best-fit result to $V_{loc}\left(0,z\right)$ (dashed black curve). Here, W=3.25a is the width of the domain along the *z*-axis used to define the B-spline basis.

**Figure 5 nanomaterials-13-02066-f005:**
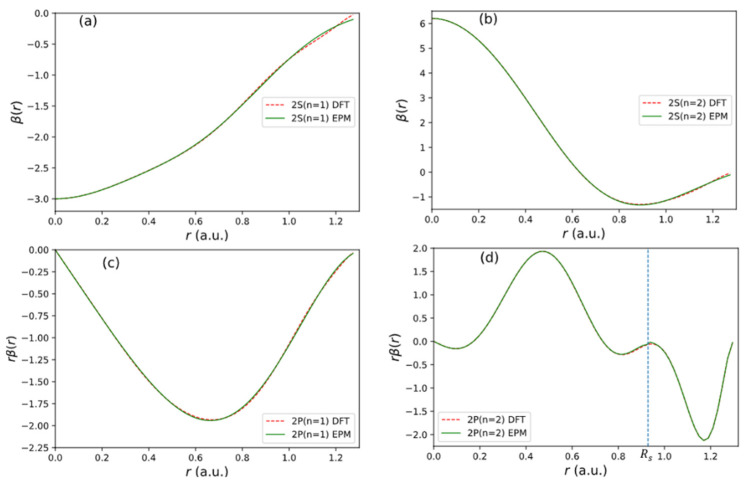
Fitting results of β functions used in the non-local pseudopotential of C atom. (**a**) 2$S_{1}$ state. (**b**) 2$S_{2}$ state. (**c**) 2$P_{1}$ state. (**d**) 2$P_{2}$ state.

**Figure 6 nanomaterials-13-02066-f006:**
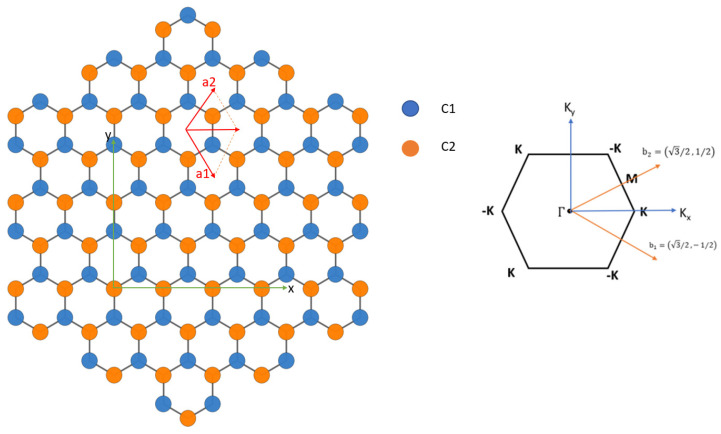
(**Left**) Primitive vectors and position of atoms for graphene. Atoms on sublattice A and B are colored blue and yellow, respectively. (**Right**) Hexagonal 2D Brillouin zone of graphene with main symmetry points. The primitive reciprocal lattice vectors shown are in units of $\frac{4\pi }{\sqrt{3}a}$.

**Figure 7 nanomaterials-13-02066-f007:**
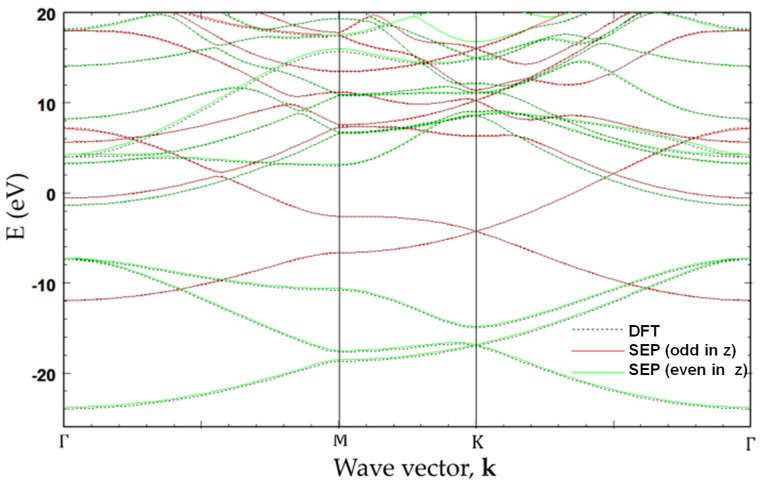
Band structure of graphene obtained by the present SEP with best-fitted parameters (solid curves). For comparison, the band structure obtained by self-consistent calculation based on DFT with Vanderbilt USPP is also included (dotted curves).

**Figure 8 nanomaterials-13-02066-f008:**
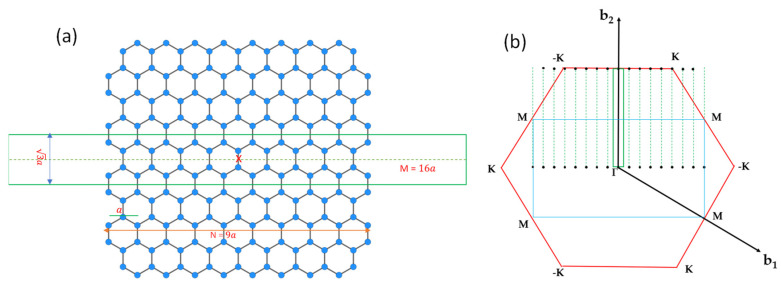
(**a**) Position of atoms of a 9 × 2 AGNR and the 16 × 2 supercell used in the calculation (enclosed within the green rectangular box). The red x marks the origin of the coordinate system to illustrate the inversion symmetry. (**b**) Rectangular 2D Brillouin zone (BZ) of the 16 × 2 supercell for the AGNRs are indicated with the green box, and the non-equivalent AGNRs reciprocal lattice vectors enclosed within the first BZ of graphene $g_{j}$
(j=1,…,2M) are indicated by black dots. The blue rectangular box indicates the BZ of a 1 × 2 supercell for graphene. The black dots outside the red hexagon (BZ of graphene) can be shifted inside the box by adding a reciprocal lattice vector.

**Figure 9 nanomaterials-13-02066-f009:**
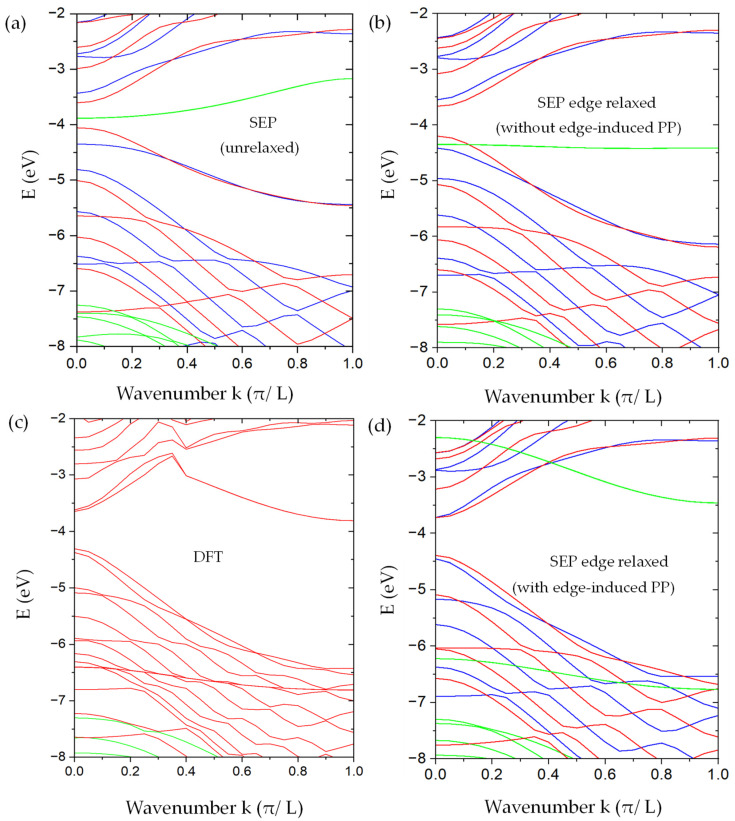
The band structure of AGNRs with M=16 and N=9. (**a**) Our SEP results without the relaxation of the edge atoms. (**b**) Our SEP results with edge relaxation but without modifying the pseudopotentials on the edge atoms. (**c**) DFT results were obtained by using the method described in [12] for the AGNRs with relaxed atomic positions for the edge atoms. (**d**) Our SEP results include the modification of pseudopotentials on the edge atoms. In SEP results (**a**,**b**,**d**), the bands in blue ($with\;\pi_{-}\;symmetry$ ) and red (with $\pi_{+}$ symmetry) are derived from π-bonded states (odd with respect to the z-mirror), while the bands in green are derived from the σ-bonded states (even with respect to the z-mirror). Here, we do not distinguish the $\sigma_{+}$ from $\sigma_{-}$ states since the important states are edge states, and they are essentially degenerate.

**Figure 10 nanomaterials-13-02066-f010:**
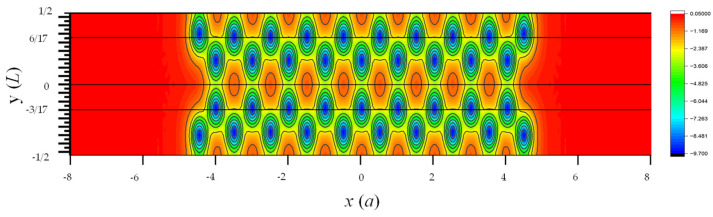
Contour plot of net local potential in one supercell of the AGNR obtained by DFT [11]. Here, a is the lattice constant of graphene, and $L=\sqrt{3}a$ is the the length of supercell along the y -axis.

**Figure 11 nanomaterials-13-02066-f011:**
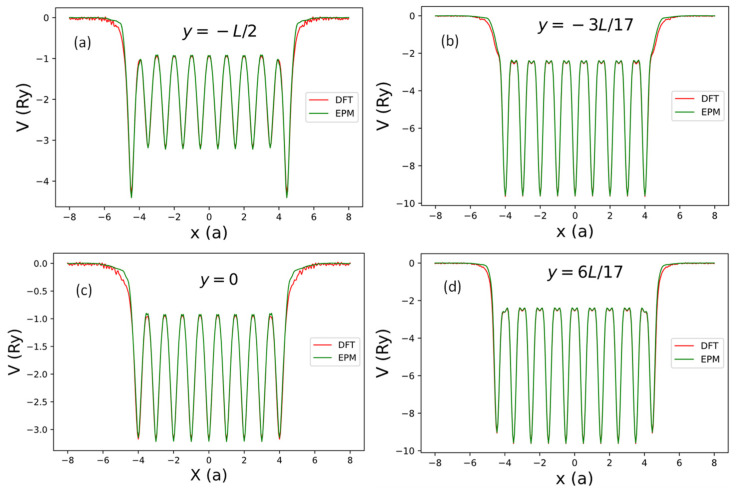
The net local pseudopotential in one supercell of the AGNRs obtained by the current SEP (green line) and by DFT (red line) along the lines with z≈0 (near the AGNRs plane) and some selected values of y. (**a**) y=−L/2 (along the bottom black line going through the centers of ten bonds in Figure 10), (**b**) y=−3L/17 (along the black line going through the centers of nine atoms in Figure 10), (**c**) y=0 (along the middle black line going through the centers of nine bonds in Figure 10), and (**d**) y=6L/17 (along the black line going through the centers of ten atoms in Figure 10).

**Figure 12 nanomaterials-13-02066-f012:**
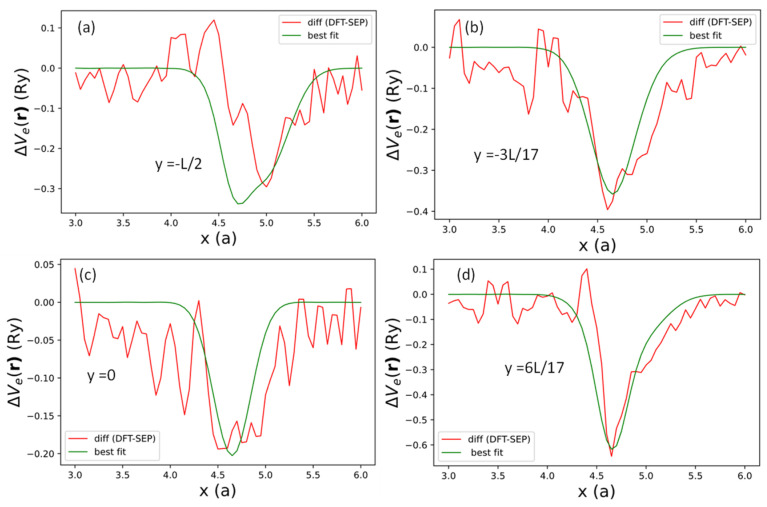
Difference in the local potentials between DFT and the current SEP results (red curves) and fitted by expression (48) (green curves) evaluated at (**a**) y=−L/2, (**b**) y=−3L/17, (**c**) y=0, and (**d**) y=6L/17, respectively.

**Figure 13 nanomaterials-13-02066-f013:**
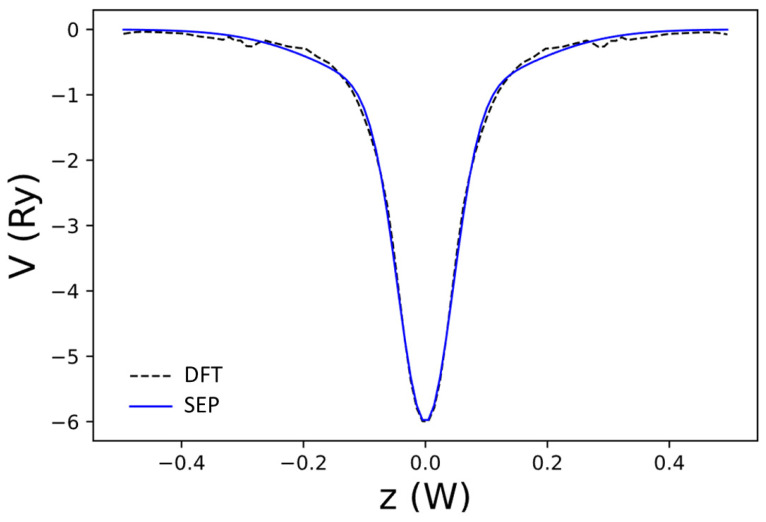
The net local potential of AGNR, $V_{loc}\left(x_{b},0,z\right),$ as a function of z obtained by DFT (dashed black curve) and the best-fit result to $V_{loc}\left(x_{b},0,z\right)$ (blue curve). Here, W=3.25a is the width of the domain along the *z*-axis used to define the B-spline basis.

**Figure 14 nanomaterials-13-02066-f014:**
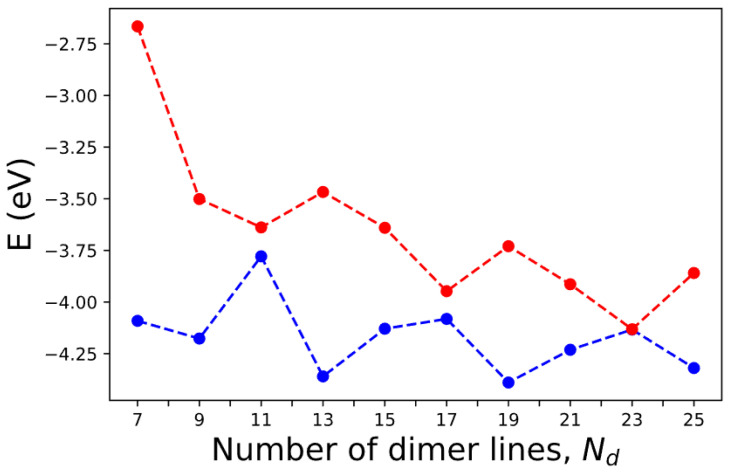
The HOMO (blue) and LUMO (red) levels of AGNRs with various numbers of dimer lines ($N_{d}$) calculated by SEPM.

**Table 1 nanomaterials-13-02066-t001:** Fitting parameters for the main part of local potential $V_{0}\left(r\right)$.

Exponents			Coefficients		
$\alpha_{1}^{}$	$\alpha_{2}^{}$	$\alpha_{3}^{}$	$C_{1}^{}$	$C_{2}^{}$	$C_{3}^{}$
0.0396	1.4100	0.3461	−0.3682	−1.7360	−1.5710

**Table 2 nanomaterials-13-02066-t002:** Fitting parameters for the short-range and long-range shape function ${f_{S}}_{}\left(z\right)$ and ${f_{L}}_{}\left(z\right)$ for the correction terms to ${\tilde{V}}_{loc}\left(z,G\right)$ used in this work.

Type	Exponent	$\mathrm{Coefficients}\;\mathrm{for}\;f_{\gamma }\left(z\right)$				$\mathrm{Coefficients}\;\mathrm{for}\;{\tilde{D}}_{\gamma }\left(G\right)$		
γ	$\alpha_{0}^{}$	$C_{1}^{\gamma }$	$C_{2}^{\gamma }$	$C_{3}^{\gamma }$	$C_{4}^{\gamma }$	$P_{1}^{\gamma }$	$P_{2}^{\gamma }$	$P_{3}^{\gamma }$
Short Range (S)	2.07	2.0372	−16.164	13.912	−2.8969	0.04494	−0.00574	0.000224
Long Range (L)	2.07	2.6251	−5.6668	2.1280	1.0239	−0.1650	0.03132	−0.002615

**Table 3 nanomaterials-13-02066-t003:** Fitting parameters for the bond-charge contribution in pseudopotential localized at the center of the hexagon cell as described by Equations (25)–(27).

Exponents in Equations (25) and (28)			Coefficients in Equation (25)					
$\alpha_{b}^{}$	$\alpha_{h1}^{}$	$\alpha_{h2}^{}$	$C_{0}^{b}$	$C_{1}^{b}$	$C_{2}^{b}$	$C_{3}^{b}$	$C_{4}^{b}$	$a_{h}$
3.0053	0.3601	0.0383	−0.1727	1.5253	−6.4817	11.5249	−5.0681	0.6930

**Table 4 nanomaterials-13-02066-t004:** Fitting parameters (a.u.) for β functions used in the non-local pseudopotential of C atom.

Orbitals	$\alpha_{}$	$R_{s}$	$C_{0}$	$C_{1}$	$C_{2}$	$C_{3}$	$C_{4}$
2$S_{1}$	2.747	1.3174	−2.999	−4.209	−11.95	7.612	0
2$S_{2}$	2.171	1.3174	6.206	−9.434	−21.03	14.21	0
2$P_{1}$	0.5104	1.3174	−3.941	−1.411	5.485	−1.939	0
2$P_{2}(seg1)$	1.134	0.9228	−2.492	94.68	−365.4	480.9	−209.8
2$P_{2}(seg2)$	0.0	1.2953	−0.9771	−350.7	4210	−5711	−4377

**Table 5 nanomaterials-13-02066-t005:** Parameters for the overlap and non-local pseudopotential terms.

n	n’	l	$E_{l}^{nn\prime }$	$q_{l}^{nn\prime }$
1	1	0	3.490422	−0.449056
1	2	0	0.207297	0.344889
2	2	0	−2.748230	−0.212785
3	3	1	2.474918	1.236379
3	4	1	−5.902130	−0.938122
4	4	1	9.289400	0.631727

**Table 6 nanomaterials-13-02066-t006:** Fitting parameters $S_{a}$, $S_{b},$ and $S_{c}$ (in a.u.) at four selected values of y.

$S_{a}$	$S_{b}$	$S_{c}$	$S_{d}$
−0.25	−0.34	0.10	−0.42

## Data Availability

Theoretical methods and results are available from the authors.

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
