# Peer review of "Semi-Empirical Pseudopotential Method for Graphene and Graphene Nanoribbons"

_nanomaterials, 2023, doi:10.3390/nano13142066_

Round 1

Reviewer 1 Report

In the present mns, Paudel et al present a semi-empirical pseudopotential (SEP) method for calculating the band structures of graphene and graphene nanoribbons. 
This approach may allow simulation of optical and transport properties of nanodevices made of graphene nanoribbons with significantly smaller computational demands.

Their study is well organized and well presented.
I think that this approach will be used by research groups who need to reduce their computer time due to large size of the calculated system.

Therefore, I recommend the publication of the mns after a minor revision.

Minor revision:

- Add empty lines between Tables or Figures and text. This will help the readers of the mns.

- Comment on the possible use of this methodology in other 2D material. 

Reviewer 2 Report

The work is of high quality and the topic important to improve calculations

of electronic properties of graphene and nanoribbons.

However, I think Nanomaterials is  not the appropiate journal to

publish these results.

Some journlas related to computational calculations should fit better the topic of this work.

Reviewer 3 Report

In the manuscript titled 'Semi-empirical Pseudopotential Method for Graphene and Graphene Nanoribbons,' written by Raj Kumar Paudel, Chung-Yuan Ren, and Yia-Chung Chang, a new semi-empirical method for calculating low-dimensional carbon materials, specifically graphene and graphene nanoribbons, is described. The main point I would like to highlight is that this paper is highly technical, providing a detailed description of a new methodology for calculating the band structures of the aforementioned materials. Given the focus on material science in MDPI Nanomaterials, it would be preferable to emphasize the material science aspects rather than the methodology of calculations.

If the Editor-in-Chief agrees with the topic (in my opinion, the methodological part of the paper should be moved to the Appendix), I would like to present the following questions and suggestions for the authors:

  1. It would be valuable to include information about the software used for the Density Functional Theory (DFT) calculations, including the basis set and functional employed. Additionally, it would be beneficial to discuss any potential software extensions that could incorporate the developed approach. Are there any other known software packages capable of performing similar calculations? This would help assess the novelty and potential advantages of the proposed method.

  2. Is there any experimental validation or verification of the calculated results, at least in part? Including some experimental data (from literature) would strengthen the reliability and applicability of the findings.

  3. In terms of practical applications, are there specific areas or industries where the proposed methodology could be utilized? Discussing potential real-world applications and their significance would enhance the relevance of the research.

  4. Considering the focus on material science in MDPI Nanomaterials, it would be beneficial to provide insights into the implications of the calculated band structures on the physical properties and behavior of graphene and graphene nanoribbons. This would help bridge the gap between the computational results and the material science perspective.

  5. Could the authors address any limitations or potential challenges associated with the proposed methodology? Discussing the constraints or possible improvements would contribute to a more comprehensive understanding of the research.

  6. Finally, I would like to add some comments about the figures and schemes. In Figure 12, please change the red and green lines to black and blue. Additionally, in Figure 8, please add the labels 'a)' and 'b)' to clarify the notations."

Round 2

Reviewer 2 Report

I have selected "Accept in present form", but I am not totally convinced...
I am still skeptic, if this work is appropiate for Nanomaterials...
Similar to the comments of reviewer 3, I find that this paper is highly technical. I do not have any doubt about the scientific quality, but I do not see any new results. This is a new methodology, it seems to be promising, but the authors do not show "new" results applying it. The achievement of results agreening with previos from other methods is the first step, but not sufficient.  I should expect the application of the new method to obtain results that are not possible with other techniques.

I let the final decision to the Editor-in-Chief, if this is appropriate or not for this journal.

Reviewer 3 Report

none. All my comments and recommendations were considered by the authors in full.